# NETWORK DECONVOLUTION

**Chengxi Ye,**[*] **Matthew Evanusa,** **Hua He,** **Anton Mitrokhin,**

**Tom Goldstein,** **James A. Yorke,**[†] **Cornelia Fermüller,** **Yiannis Aloimonos**

Department of Computer Science, University of Maryland, College Park
`{cxy, mevanusa, huah, amitrokh}@umd.edu`
`{tomg@cs,yorke@,fer@umiacs,yiannis@cs}.umd.edu`

## ABSTRACT

Convolution is a central operation in Convolutional Neural Networks (CNNs), which applies a kernel to overlapping regions shifted across the image. However, because of the strong correlations in real-world image data, convolutional kernels are in effect re-learning redundant data. In this work, we show that this redundancy has made neural network training challenging, and propose *network deconvolution*, a procedure which optimally removes pixel-wise and channel-wise correlations before the data is fed into each layer. Network deconvolution can be efficiently calculated at a fraction of the computational cost of a convolution layer. We also show that the deconvolution filters in the first layer of the network resemble the center-surround structure found in biological neurons in the visual regions of the brain. Filtering with such kernels results in a sparse representation, a desired property that has been missing in the training of neural networks. Learning from the sparse representation promotes faster convergence and superior results *without* the use of batch normalization. We apply our network deconvolution operation to 10 modern neural network models by replacing batch normalization within each. Extensive experiments show that the network deconvolution operation is able to deliver performance improvement in all cases on the CIFAR-10, CIFAR-100, MNIST, Fashion-MNIST, Cityscapes, and ImageNet datasets.

## 1 INTRODUCTION

Images of natural scenes that the human eye or camera captures contain adjacent pixels that are statistically highly correlated (Olshausen & Field, 1996; Hyvrinen et al., 2009), which can be compared to the correlations introduced by *blurring* an image with a Gaussian kernel. We can think of images as being convolved by an unknown filter (Figure 1). The correlation effect complicates object recognition tasks and makes neural network training challenging, as adjacent pixels contain redundant information.

It has been discovered that there exists a visual correlation removal processes in animal brains. Visual neurons called retinal ganglion cells and lateral geniculate nucleus cells have developed "Mexican hat"-like circular center-surround receptive field structures to reduce visual information redundancy, as found in Hubel and Wiesel's famous cat experiment (Hubel & Wiesel, 1961; 1962). Furthermore, it has been argued that data compression is an essential and fundamental processing step in natural brains, which inherently involves removing redundant information and only keeping the most salient features (Richert et al., 2016).

In this work, we introduce *network deconvolution*, a method to reduce redundant correlation in images. Mathematically speaking, a correlated signal is generated by a convolution: $b = k * x = Kx$ (as illustrated in Fig. 1 right), where $k$ is the kernel and $K$ is the corresponding convolution matrix. The purpose of network deconvolution is to remove the correlation effects via: $x = K^{-1}b$, assuming $K$ is an invertible matrix.

---

[*]Corresponding Author
[†]Institute for Physical Science and Technology

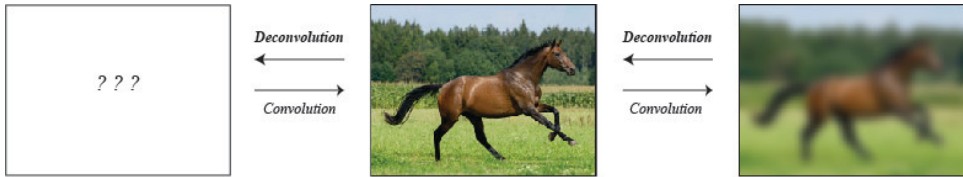

Figure 1: Performing convolution on this real world image using a correlative filter, such as a Gaussian kernel, adds correlations to the resulting image, which makes object recognition more difficult. The process of removing this blur is called *deconvolution*. What if, however, what we saw as the real world image was *itself* the result of some unknown correlative filter, which has made recognition more difficult? Our proposed network deconvolution operation can decorrelate underlying image features which allows neural networks to perform better.

Image data being fed into a convolutional network (CNN) exhibit two types of correlations. Neighboring pixels in a single image or feature map have high *pixel-wise correlation*. Similarly, in the case of different channels of a hidden layer of the network, there is a strong correlation or "cross-talk" between these channels; we refer to this as *channel-wise correlation*. The goal of this paper is to show that both kinds of correlation or redundancy hamper effective learning. Our *network deconvolution* attempts to remove both correlations in the data at every layer of a network.

Our contributions are the following:

- We introduce *network deconvolution*, a decorrelation method to remove both the pixel-wise and channel-wise correlation at each layer of the network.
- Our experiments show that deconvolution can replace batch normalization as a generic procedure in a variety of modern neural network architectures with better model training.
- We prove that this method is the optimal transform if considering $L_2$ optimization.
- Deconvolution has been a misnomer in convolution architectures. We demonstrate that network deconvolution is indeed a *deconvolution* operation.
- We show that network deconvolution reduces redundancy in the data, leading to sparse representations at each layer of the network.
- We propose a novel implicit deconvolution and subsampling based acceleration technique allowing the deconvolution operation to be done at a cost fractional to the corresponding convolution layer.
- We demonstrate the network deconvolution operation improves performance in comparison to batch normalization on the CIFAR-10, CIFAR-100, MNIST, Fashion-MNIST, Cityscapes, and ImageNet datasets, using 10 modern neural network models.

## 2 RELATED WORK

### 2.1 NORMALIZATION AND WHITENING

Since its introduction, batch normalization has been the main normalization technique (Ioffe & Szegedy, 2015) to facilitate the training of deep networks using stochastic gradient descent (SGD). Many techniques have been introduced to address cases for which batch normalization does not perform well. These include training with a small batch size (Wu & He, 2018), and on recurrent networks (Salimans & Kingma, 2016). However, to our best knowledge, none of these methods has demonstrated improved performance on the ImageNet dataset.

In the signal processing community, our network deconvolution could be referred to as a *whitening deconvolution*. There have been multiple complicated attempts to whiten the feature channels and to utilize second-order information. For example, authors of (Martens & Grosse, 2015; Desjardins et al., 2015) approximate second-order information using the Fisher information matrix. There, the whitening is carried out interwoven into the back propagation training process.

The correlation between feature channels has recently been found to hamper the learning. Simple approximate whitening can be achieved by removing the correlation in the channels. Channel-wise

decorrelation was first proposed in the backward fashion (Ye et al., 2017). Equivalently, this procedure can also be done in the forward fashion by a change of coordinates (Ye et al., 2018; Huang et al., 2018; 2019). However, none of these methods has captured the nature of the *convolution* operation, which specifically deals with the *pixels*. Instead, these techniques are most appropriate for the standard linear transform layers in fully-connected networks.

## 2.2 DECONVOLUTION OF DNA SEQUENCING SIGNALS

Similar correlation issues also exist in the context of DNA sequencing (Ye et al., 2014) where many DNA sequencers start with analyzing correlated signals. There is a cross-talk effect between different sensor channels, and signal responses of one nucleotide base spread into its previous and next nucleotide bases. As a result the sequencing signals display both channel correlation and pixel correlation. A blind deconvolution technique was developed to estimate and remove kernels to recover the unblurred signal.

## 3 MOTIVATIONS

### 3.1 SUBOPTIMALITY OF EXISTING TRAINING METHODS

Deep neural network training has been a challenging research topic for decades. Over the past decade, with the regained popularity of deep neural networks, many techniques have been introduced to improve the training process. However, most of these methods are sub-optimal even for the most basic linear regression problems.

Assume we are given a linear regression problem with $L_2$ loss (Eq. 1). In a typical setting, the output $y = Xw$ is given by multiplying the inputs $X$ with an unknown weight matrix $w$, which we are solving for. In our paper, with a slight abuse of notation, $X$ can be the data matrix or the augmented data matrix $(X|1)$.

$$Loss_{L_2} = \frac{1}{2}\|y - \hat{y}\|^2 = \frac{1}{2}\|Xw - \hat{y}\|^2. \tag{1}$$

Here $\hat{y}$ is the response data to be regressed. With neural networks, gradient descent iterations (and its variants) are used to solve the above problem. To conduct one iteration of gradient descent on Eq. 1, we have:

$$w_{new} = w_{old} - \alpha\frac{1}{N}(X^t X w_{old} - X^t\hat{y}). \tag{2}$$

Here $\alpha$ is the step length or learning rate. Basic numerical experiments tell us these iterations can take long to converge. Normalization techniques, which are popular in training neural networks, are beneficial, but are generally not optimal for this simple problem. If methods are suboptimal for simplest linear problems, it is less likely for them to be optimal for more general problems. Our motivation is to find and apply what is linearly optimal to the more challenging problem of network training.

For the $L_2$ regression problem, an optimal solution can be found by setting the gradient to 0: $\frac{\partial Loss_{L_2}}{\partial w} = X^t(Xw - \hat{y}) = 0$

$$w = (X^t X)^{-1} X^t\hat{y} \tag{3}$$

Here, we ask a fundamental question: When can gradient descent converge in *one single iteration*?

**Proposition 1.** *Gradient descent converges to the optimal solution in one iteration if $\frac{1}{N}X^t X = I$.*

*Proof.* Substituting $\frac{1}{N}X^t X = I$, into the optimal solution (Eq. 3) we have $w = \frac{1}{N}X^t\hat{y}$.

On the other hand, substituting the same condition with $\alpha = 1$ in Eq. 2 we have $w_{new} = \frac{1}{N}X^t\hat{y}$. □

Since gradient descent converges in one single step, the above proof gives us the optimality condition.

$\frac{1}{N}X^tX = I$ calculates the covariance matrix of the features. The optimal condition suggests that the features should be standardized and uncorrelated with each other. When this condition does not hold, the gradient direction does not point to the optimal solution. In fact, the more correlated the data, the slower the convergence (Richardson, 1911). This problem could be handled equivalently (Section A.8) either by correcting the gradient by multiplying the $Hessian$ matrix $= \frac{1}{N}(X^tX)^{-1}$, or by a change of coordinates so that in the new space we have $\frac{1}{N}X^tX = I$. This paper applies the latter method for training convolutional networks.

## 3.2 Need of Support for Convolutions

Even though normalization methods were developed for training convolutional networks and have been found successful, these methods are more suitable for non-convolutional operations. Existing methods normalize features by channel or by layer, irrespective of whether the underlying operation is convolutional or not. We will show in section 4.1 that if the underlying operation is a convolution, this usually implies a strong violation of the optimality condition.

## 3.3 A Neurological Basis for Deconvolution

Many receptive fields in the primate visual cortex exhibit center-surround type behavior. Some receptive fields, called on-center cells respond maximally when a stimuli is *given* at the center of the receptive field and a *lack* of stimuli is given in a circle surrounding it. Some others, called off-center cells respond maximally in the reversed way when a lack of stimuli is at the center and the stimuli is given in a circle surrounding it (Fig. 9) (Hubel & Wiesel, 1961; 1962). It is well-understood that these center-surround fields form the basis of the simple cells in the primate V1 cortex.

If the center-surround structures are beneficial for learning, one might expect such structures to be learned in the network training process. As shown in the proof above, the minima is the same with or without such structures, so gradient descent does not get an incentive to develop a faster solution, rendering the need to develop such structures externally. As shown later in Fig. 2, our deconvolution kernels strongly resemble center-surround filters like those in nature.

## 4 The Deconvolution Operation

### 4.1 The Matrix Representation of a Convolution Layer

The standard convolution filtering $x * kernel$, can be formulated into one large matrix multiplication $Xw$ (Fig. 3). In the 2-dimensional case, $w$ is the flattened 2D $kernel$. The first column of $X$ corresponds to the flattened image patch of $x[1 : H - k, 1 : W - k]$, where $k$ is the side length of the kernel. Neighboring columns correspond to shifted patches of $x$: $X[:, 2] = vec(x[1 : H - k, 2 : W - k + 1]), ..., X[:, k^2] = vec(x[k : H, k : W])$. A commonly used function $im2col$ has been designed for this operation. Since the columns of $X$ are constructed by shifting large patches of $x$ by one pixel, the columns of $X$ are heavily correlated with each other, which strongly violates the optimality condition. This violation slows down the training algorithm (Richardson, 1911), and cannot be addressed by normalization methods (Ioffe & Szegedy, 2015).

For a regular convolution layer in a network, we generally have multiple input feature channels and multiple kernels in a layer. We call $im2col$ in each channel, and horizontally concatenate the resulting data matrices from each individual channel to construct the full data matrix, then vectorize and concatenate all the kernels to get $w$. Matrix vector multiplication is used to calculate the output $y$, which is then reshaped into the output shape of the layer. Similar constructions can also be developed for the specific convolution layers such as the grouped convolution, where we carry out such constructions for each group. Other scenarios such as when the group number equals the channel number (channel-wise conv) or when $k = 1$ ($1 \times 1$ conv) can be considered as special cases.

In Fig. 3 (top right) we show as an illustrative example the resulting calculated covariance matrix of a sample data matrix $X$ in the first layer of a VGG network (Simonyan & Zisserman, 2014) taken from one of our experiments. The first layer is a $3 \times 3$ convolution that mixes RGB channels. The total dimension of the weights is 27, the corresponding covariance matrix is $27 \times 27$. The diagonal blocks correspond to the pixel-wise correlation within $3 \times 3$ neighborhoods. The off diagonal blocks

correspond to correlation of pixels across different channels. We have empirically seen that natural images demonstrate stronger pixel-wise correlation than cross-channel correlation, as the diagonal blocks are brighter than the off diagonal blocks.

## 4.2 THE DECONVOLUTION OPERATION

Once the covariance matrix has been calculated, an inverse correction can be applied. It is beneficial to conduct the correction in the forward way for numerical accuracy and because the gradient of the correction can also be included in the gradient descent training.

Given a data matrix $X_{N \times F}$ as described above in section 4.1, where $N$ is the number of samples, and $F$ is the number of features, we calculate the covariance matrix $Cov = \frac{1}{N}(X - \mu)^T(X - \mu)$.

We then calculate an approximated inverse square root of the covariance matrix $D = Cov^{-\frac{1}{2}}$ (see section 4.5.3) and multiply this with the centered vectors $(X - \mu) \cdot D$. In a sense, we remove the correlation effects both pixel-wise and channel-wise. If computed perfectly, the transformed data has the identity matrix as covariance: $D^T(X - \mu)^T(X - \mu)D = Cov^{-0.5} \cdot Cov \cdot Cov^{-0.5} = I$.

Algorithm 1 describes the process to construct $X$ and $D \approx (Cov + \epsilon \cdot I)^{-\frac{1}{2}}$. Here $\epsilon \cdot I$ is introduced to improve stability. We then apply the deconvolution operation via matrix multiplication to remove the correlation between neighboring pixels and across different channels. The deconvolved data is then multiplied with $w$. The full equation becomes $y = (X - \mu) \cdot D \cdot w + b$, or simply $y = X \cdot D \cdot w$ if $X$ is the augmented data matrix (Fig. 3).

We denote the deconvolution operation in the $i$-th layer as $D_i$. Hence, the input to next layer $x_{i+1}$ is:

$$x_{i+1} = f_i \circ W_i \circ D_i \circ x_i, \tag{4}$$

where $\circ$ is the (right) matrix multiplication operation, $x_i$ is the input coming from the $i-$th layer, $D_i$ is the deconvolution operation on that input, $W_i$ is the weights in the layer, and $f_i$ is the activation function.

## 4.3 JUSTIFICATIONS

### 4.3.1 ON NAMING THE METHOD DECONVOLUTION

The name network deconvolution has also been used in inverting the convolution effects in biological networks (Feizi et al., 2013). We prove that our operation is indeed a generalized *deconvolution* operation.

**Proposition 2.** *Removal of pixel-wise correlation (or patch-based whitening) is a deconvolution operation.*

*Proof.* Let $\delta$ be the delta kernel, $x$ be an arbitrary signal, and $X = im2col(x)$.

$$x = x * \delta = X \cdot \delta = X \cdot Cov^{-0.5} \cdot Cov^{0.5} \cdot \delta = X \cdot Cov^{-0.5} \cdot k_{cov} = X \cdot \delta = x \tag{5}$$

The above equations show that the deconvolution operation negates the effects of the convolution using kernel $k_{cov} = Cov^{0.5} \cdot \delta$. $\square$

### 4.3.2 THE DECONVOLUTION KERNEL

The deconvolution kernel can be found as $Cov^{-0.5} \cdot vec(\delta)$, where $vec(:)$ is the Vectorize function, or equivalently by slicing the middle row/column of $Cov^{-0.5}$ and reshaping it into the kernel size. We visualize the deconvolution kernel from $1024$ random images from the ImageNet dataset. The kernels indeed show center-surround structure, which coincides with the biological observation (Fig. 2). The filter in the green channel is an on-center cell while the other two are off-center cells (Hubel & Wiesel, 1961; 1962).

## 4.4 OPTIMALITY

Motivated by the ubiquity of center-surround and lateral-inhibition mechanisms in biological neural systems, we now ask if removal of redundant information, in a manner like our network deconvolution, is an optimal procedure for learning in neural networks.

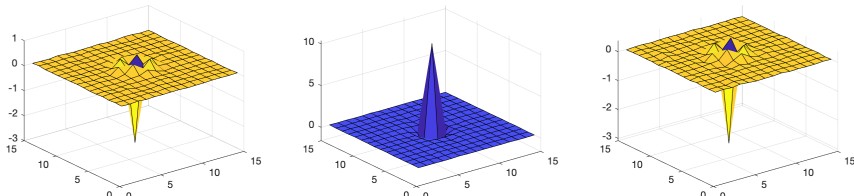

Figure 2: Visualizing the $15 \times 15$ deconvolution kernels from 1024 random images from the ImageNet dataset. The kernels in the R,G,B channels consistently show center-surround structures.

### 4.4.1 $L_2$ OPTIMIZATIONS

There is a classic kernel estimation problem (Cho & Lee, 2009; Ye et al., 2014) that requires solving for the kernel given the input data $X$ and the blurred output data $y$. Because $X$ violates the optimality condition, it takes tens or hundreds of gradient descent iterations to converge to a close enough solution. We have demonstrated that our deconvolution processing is optimal for the kernel estimation, in contrast to all other normalization methods.

### 4.4.2 ON TRAINING NEURAL NETWORKS

Training convolutional neural networks is analogous to a series of kernel estimation problem, where we have to solve for the kernels in each layer. Network deconvolution has a favorable near-optimal property for training neural networks.

For simplicity, we assume the activation function is a sample-variant matrix multiplication throughout the network. The popular $ReLU$ (Nair & Hinton, 2010) activation falls into this category. Let $W$ be the linear transform/convolution in a certain layer, $A$ the inputs to the layer, $B$ the operation from the output of the current layer to the output of the last deconvolution operation in the network. The computation of such a network can be formulated as: $y = AWB$.

**Proposition 3.** *Network deconvolution is near-optimal if $W$ is an orthogonal transform and if we connect the output $y$ to the $L_2$ loss.*

Rewriting the matrix equation from the previous subsection using the Kronecker product, we get: $y = AWB = (B^T \otimes A)vec(W) = Xvec(W)$, where $\otimes$ is the Kronecker product. According to the discussion in the previous subsection, gradient descent is optimal if $X = (B^T \otimes A)$ satisfies the orthogonality condition. In a network trained with deconvolution, the input $A$ is orthogonal for each batch of data. If $B$ is also orthogonal, then according to basic properties of the Kronecker product (Golub & van Loan, 2013)(Ch 12.3.1), $(B^T \otimes A)$ is also orthogonal. $y$ is orthogonal since it is the output of the last deconvolution operation in our network. If we assume $W$ is an orthogonal transform, then $B$ transforms orthogonal inputs to orthogonal outputs, and is approximately orthogonal.

Slight loss of optimality incurs since we do not enforce $W$ to be orthogonal. But the gain here is that the network is unrestricted and is promised to be as powerful as any standard network. On the other hand, it is worth mentioning that many practical loss functions such as the cross entropy loss have similar shapes to the $L_2$ loss.

### 4.5 ACCELERATIONS

We note that in a direct implementation, the runtime of our training using deconvolution is slower than convolution using the wallclock as a metric. This is due to the suboptimal support in the implicit calculation of the matrices in existing libraries. We propose acceleration techniques to reduce the deconvolution cost to only a fraction of the convolution layer (Section A.6). Without further optimization, our training speed is similar to training a network using batch normalization on the ImageNet dataset while achieving better accuracy. This is a desired property when faced with difficult models (Goodfellow et al., 2014) and with problems where the network part is not the major bottleneck (Ye et al., 2018).

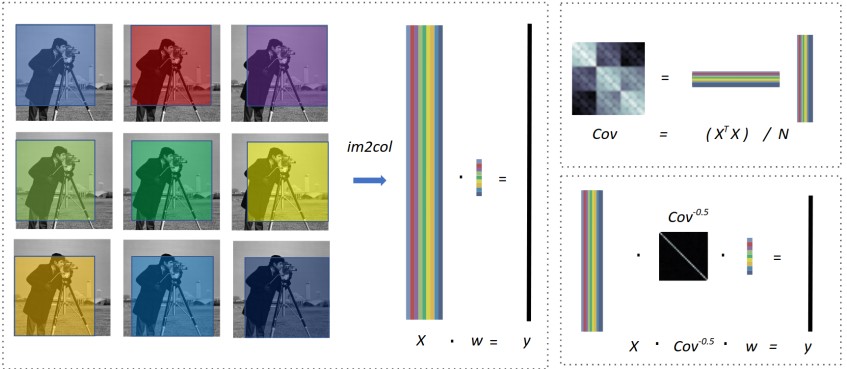

Figure 3: (Left) Given a single channel image, and a $3 \times 3$ kernel, the kernel is first flattened into a 9 dimensional vector $w$. The 9 image patches, corresponding to the image regions each kernel entry sees when overlaying the kernel over the image and then shifting the kernel one pixel each step, are flattened into a tall matrix $X$. It is important to note that because the patches are shifted by just one pixel, the columns of $X$ are highly correlated. The output $y$ is calculated with matrix multiplication $Xw$, which is then reshaped back into a $2D$ image. (Top Right) In a convolution layer the matrix $X$ and $Cov$ is calculated from Algorithm 1. (Bottom Right) The pixel-wise and channel-wise correlation is removed by multiplying this X matrix with with $Cov^{-\frac{1}{2}}$, before the weight training.

### 4.5.1 Implicit Deconvolution

Following from the associative rule of matrix multiplication, $y = X \cdot D \cdot w = X \cdot (D \cdot w)$, which suggests that the deconvolution can be carried out *implicitly* by changing the model parameters, without explicitly deconvolving the data. Once we finish the training, we freeze $D$ to be the running average. This change of parameters makes a network with deconvolution perform faster at testing time, which is a favorable property for mobile applications. We provide the practical recipe to include the bias term: $y = (X - \mu) \cdot D \cdot w + b = X \cdot (D \cdot w) + b - \mu \cdot D \cdot w$.

### 4.5.2 Fast Computation of the Covariance Matrice

We propose a simple $S = 3 - 5\times$ subsampling technique that speeds up the computation of the covariance matrix by a factor of $10 - 20$. Since the number of involved pixels is usually large compared with the degree of freedom in a convariance matrix (Section A.6), this simple strategy provides significant speedups while maintaining the training quality. Thanks to the regularization and the iterative method we discuss below, we found the subsampling method to be robust even when the covariance matrix is large.

### 4.5.3 Fast Inverse Square Root of the Covariance Matrix

Computing the inverse square root has a long and fruitful history in computer graphics and numerical mathematics. Fast computation of the inverse square root of a scalar with Newton-Schulz iterations has received wide attention in game engines (Lomont, 2003). One would expect the same method to seamlessly generalize to the matrix case. However, according to numerous experiments, the standard Newton-Schulz iterations suffer from severe numerical instability and explosion after $\sim 20$ iterations for simple matrices (Section A.3) (Higham, 1986)(Eq. 7.12), (Higham, 2008). Coupled Newton-Schulz iterations have been designed (Denman & Beavers, 1976) (Eq.6.35), (Higham, 2008) and been proved to be numerically stable.

We compute the approximate inverse square root of the covariance matrix at low cost using coupled Newton-Schulz iteration, inspired by the Denman-Beavers iteration method (Denman & Beavers, 1976). Given a symmetric positive definite covariance matrix $Cov$, the coupled Newton-Schulz iterations start with initial values $Y_0 = Cov$, $Z_0 = I$. The iteration is defined as: $Y_{k+1} = \frac{1}{2}Y_k(3I - Z_kY_k)$, $Z_{k+1} = \frac{1}{2}(3I - Z_kY_k)Z_k$, and $Y_k \rightarrow Cov^{\frac{1}{2}}$, $Z_k \rightarrow Cov^{-\frac{1}{2}}$ (Higham, 2008) (Eq.6.35). Note that this coupled iteration has been used in recent works to calculate the square root of a matrix (Lin & Maji, 2017). Instead, we take the *inverse* square root from the outputs, as first shown in (Ye et al.,

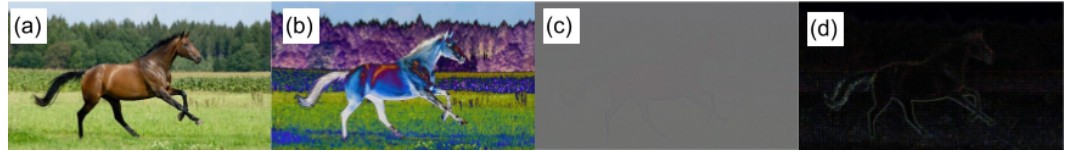

Figure 4: (a) The input image. (b) The absolute value of the zero-meaned input image. (c) The deconvolved input image (min-max normalized, gray areas stands for 0). (d) Taking the absolute value of the deconvolved image.

2018). In contrast with the vanilla Newton-Schulz method (Higham, 2008; Huang et al., 2019)(Eq. 7.12), we found the coupled Newton-Schulz iterations are stable even if iterated for thousands of times.

It is important to point out a practical implementation detail: when we have $Ch_{in}$ input feature channels, and the kernel size is $k \times k$, the size of the covariance matrix is $(Ch_{in} \times k \times k) \times (Ch_{in} \times k \times k)$. The covariance matrix becomes large in deeper layers of the network, and inverting such a matrix is cumbersome. We take a grouping approach by evenly dividing the feature channels $Ch_{in}$ into smaller blocks (Ye et al., 2017; Wu & He, 2018; Ye et al., 2018); we us $b$ to denote the block size, and usually set $B = 64$. The mini-batch covariance of a each block has a manageable size of $(B \times k \times k) \times (B \times k \times k)$. Newton-Schulz iterations are therefore conducted on smaller matrices. We notice that only a few ($\sim 5$) iterations are necessary to achieve good performance. Solving for the inverse square root takes $O((k \times k \times B)^3)$. The computation of the covariance matrix has complexity $O(H \times W \times k \times k \times B \times B \times \frac{Ch_{in}}{B} \times \frac{1}{S \times S}) = O(H \times W \times k \times k \times Ch_{in} \times B \times \frac{1}{S \times S})$. Implicit deconvolution is a simple matrix multiplication with complextity $O(Ch_{out} \times (B \times k \times k)^2 \times \frac{Ch_{in}}{B})$. The overall complexity is $O(\frac{H \times W \times k \times k \times Ch_{in} \times B}{S \times S} + (k \times k \times B)^3 + Ch_{out} \times (B \times k \times k)^2 \times \frac{Ch_{in}}{B})$, which is usually a small fraction of the cost of the convolution operation (Section A.6). In comparison, the computational complexity of a regular convolution layer has a complexity of $O(H \times W \times k \times k \times Ch_{in} \times Ch_{out})$.

---

**Algorithm 1** Computing the Deconvolution Matrix

---

1: **Input:** C channels of input features $[x_1, x_2, ..., x_C]$
2: **for** $i \in \{1, ..., C\}$ **do**
3:    $X_i = im2col(x_i)$
4: **end for**
5: $X = [X_1, ..., X_C]$ %Horizontally Concatenate
6: $X = Reshape(X)$ %Divide columns into groups
7: $Cov = \frac{1}{N} X^t X$
8: $D \approx (Cov + \epsilon \cdot I)^{-\frac{1}{2}}$

---

## 4.6 SPARSE REPRESENTATIONS

Our deconvolution applied at each layer removes the pixel-wise and channel-wise correlation and transforms the original dense representations into sparse representations (in terms of heavy-tailed distributions) without losing information. This is a desired property and there is a whole field with wide applications developed around sparse representations (Hyvrinen et al., 2009)(Fig. 7.7), (Olshausen & Field, 1996; Ye et al., 2013). In Fig. 4, we visualize the deconvolution operation on an input and show how the resulting representations ( 4(d)) are much sparser than the normalized image ( 4(b)). We randomly sample 1024 images from the ImageNet and plot the histograms and log density functions before and after deconvolution (Fig. 10). After deconvolution, the log density distribution becomes heavy-tailed. This holds true also for hidden layer representations (Section A.5). We show in the supplementary material (Section A.4) that the sparse representation makes classic regularizations more effective.

## 5 A UNIFIED VIEW

Network deconvolution is a forward correction method that has relations to several successful techniques in training neural networks. When we set $k = 1$, the method becomes channel-wise decorrelation, as in (Ye et al., 2018; Huang et al., 2018). When $k = 1, B = 1$, network deconvolution is Batch Normalization (Ioffe & Szegedy, 2015). If we apply the decorrelation in a backward way in the gradient direction, network deconvolution is similar to $SGD2$ (Ye et al., 2017), natural gradient descent (Desjardins et al., 2015) and $KFAC$ (Martens & Grosse, 2015), while being more efficient and having better numerical properties (Section A.8).

## 6 EXPERIMENTS

We now describe experimental results validating that network deconvolution is a powerful and successful tool for sharpening the data. Our experiments show that it outperforms identical networks using batch normalization (Ioffe & Szegedy, 2015), a major method for training neural networks. As we will see across all experiments, deconvolution not only improves the final accuracy but also decreases the amount of iterations it takes to learn a reasonably good set of weights in a small number of epochs.

**Linear Regression with $L_2$ loss and Logistic Regression:** As a first experiment, we ran network deconvolution on a simple linear regression task to show its efficacy. We select the Fashion-MNIST dataset. It is noteworthy that with binary targets and the $L_2$ loss, the problem has an explicit solution if we feed the whole dataset as input. This problem is the classic kernel estimation problem, where we need to solve for 10 optimal $28 \times 28$ kernels to convolve with the inputs and minimize the $L_2$ loss for binary targets. During our experiment, we notice that it is important to use a small learning rate of $0.02 - 0.1$ for vanilla $SGD$ training to prevent divergence. However, we notice that with deconvolution we can use the optimal learning rate $1.0$ and get high accuracy as well. It takes $\sim 5$ iterations to get to a low cost under the mini-batch setting (Fig. 5(a)). This even holds if we change the loss to logistic regression loss (Fig. 5(b)).

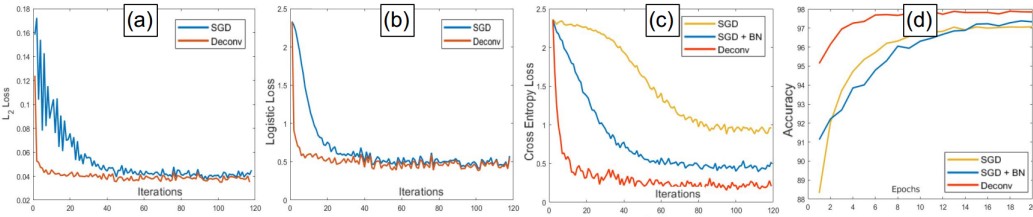

Figure 5: (a-b): Regression losses on Fashion-MNIST dataset showing the effectiveness of deconvolution versus batch normalization on a non-convolutional type layer. (a) One layer, linear regression model with $L_2$ loss. (b) One layer, linear regression model with logistic regression loss. (c-d): Results of a 3-hidden-layer Multi Layer Perceptron (MLP) network on the MNIST dataset.

**Convolutional Networks on CIFAR-10/100:** We ran deconvolution on the CIFAR-10 and CIFAR-100 datasets (Table 1), where we compared again the use of network deconvolution versus the use of batch normalization. Across 10 modern network architectures for both datasets, deconvolution consistently improves convergence on these well-known datasets. There is a wide performance gap after the first epochs of training. Deconvolution leads to faster convergence: 20-epoch training using deconvolution leads to results that are comparable to 100-epoch training using batch normalization.

In our setting, we remove all batch normalizations in the networks and replace them with deconvolution before each convolution/fully-connected layer. For the convolutional layers, we set $B = 64$ before calculating the covariance matrix. For the fully-connected layers, we set $B$ equal to the input channel number, which is usually $512$. We set the batch size to $128$ and the weight decay to $0.001$. All models are trained with $SGD$ and a learning rate of $0.1$.

**Convolutional Networks on ImageNet:** We tested three widely acknowledged model architectures (VGG-11, ResNet-18, DenseNet-121) from the PyTorch model zoo and find significant improvements on both networks over the reference models. Notably, for the VGG-11 network, we notice our method

| | | CIFAR-10 | | | | | | CIFAR-100 | | | | | |
|---|---|---|---|---|---|---|---|---|---|---|---|---|---|
| | Net Size | BN 1 | ND 1 | BN 20 | ND 20 | BN 100 | ND 100 | BN 1 | ND 1 | BN 20 | ND 20 | BN 100 | ND 100 |
| VGG-16 | 14.71M | 14.12% | **74.18%** | 90.07% | **93.25%** | 93.58% | **94.56%** | 2.01% | **37.94%** | 63.22% | **71.97%** | 72.75% | **75.32%** |
| ResNet-18 | 11.17M | 56.25% | **72.89%** | 92.64% | **94.07%** | 94.87% | **95.40%** | 16.10% | **35.73%** | 72.67% | **76.55%** | 77.70% | **78.63%** |
| Preact-18 | 11.17M | 55.15% | **72.70%** | 91.93% | **94.10%** | 94.37% | **95.44%** | 15.17% | **36.52%** | 70.79% | **76.04%** | 76.14% | **79.14%** |
| DenseNet-121 | 6.88M | 59.56% | **76.63%** | 93.25% | **94.89%** | 94.71% | **95.88%** | 17.90% | **42.91%** | 74.79% | **77.63%** | 77.99% | **80.69%** |
| ResNext-29 | 4.76M | 52.14% | **69.22%** | 93.12% | **94.05%** | 95.15% | **95.80%** | 17.98% | **30.93%** | 74.26% | **77.35%** | 78.60% | **80.34%** |
| MobileNet v2 | 2.28M | 54.29% | **65.40%** | 89.86% | **92.52%** | 90.51% | **94.35%** | 15.88% | **29.01%** | 66.31% | **72.33%** | 67.52% | **74.90%** |
| DPN-92 | 34.18M | 34.00% | **53.02%** | 92.87% | **93.74%** | 95.14% | **95.82%** | 8.84% | **21.89%** | 74.87% | **76.12%** | 78.87% | **80.38%** |
| PNASNetA | 0.13M | 21.81% | **64.19%** | 75.85% | **81.97%** | 81.22% | **84.45%** | 10.49% | **36.52%** | 44.60% | **55.65%** | 54.52% | **59.44%** |
| SENet-18 | 11.26M | 57.63% | **67.21%** | 92.37% | **94.11%** | 94.57% | **95.38%** | 16.60% | **32.22%** | 71.10% | **75.79%** | 76.41% | **78.63%** |
| EfficientNet | 2.91M | 35.40% | **55.67%** | 84.21% | **86.78%** | 86.07% | **88.42%** | 19.03% | **22.40%** | 57.23% | **57.59%** | 59.09% | **62.37%** |

Table 1: Comparison on CIFAR-10/100 over 10 modern CNN architectures. Models are trained for 1, 20, 100 epochs using batch normalization (BN) and network deconvolution (ND). Every single model shows improved accuracy using network deconvolution.

| | VGG-11 | | | ResNet-18 | | DenseNet-121 | |
|---|---|---|---|---|---|---|---|
| | Original | BN | Deconv | BN | Deconv | BN | Deconv |
| ImageNet top-1 | 69.02% | 70.38% | **71.95%** | 69.76% | **71.24%** | 74.65% | **75.73%** |
| ImageNet top-5 | 88.63% | 89.81% | **90.49%** | 89.08% | **90.14%** | 92.17% | **92.75%** |

Table 2: Comparison of accuracies of deconvolution with the model zoo implementation of VGG-11, ResNet-18, DenseNet-121 on ImageNet with batch normalization using the reference implementation on PyTorch. For VGG-11, we also include the performance of the original network without batch normalization.

has led to significant improved accuracy. The top-1 accuracy is even higher than $71.55\%$, reported by the reference VGG-13 model trained with batch normalization. The improvement introduced by network deconvolution is twice as large as that from batch normalization ($+1.36\%$). This fact also suggests that improving the training method may be more effective than improving the architecture.

We keep most of the default settings when training the models. We set $B = 64$ for all deconvolution operations. The networks are trained for 90 epochs with a batch size of 256, and weight decay of 0.0001. The initial learning rates are 0.01, 0.1 and 0.1, respectively for VGG-11, ResNet-18 and DenseNet-121 as described in the paper. We used cosine annealing to smoothly decrease the learning rate to compare the curves.

**Generalization to Other Tasks** It is worth pointing out that network deconvolution can be applied to other tasks that have convolution layers. Further results on semantic segmentation on the Cityscapes dataset can be found in (Sec. A.8). Also, the same deconvolution procedure for $1 \times 1$ convolutions can be used for non-convolutional layers, which makes it useful for the broader machine learning community. We constructed a 3-layer fully-connected network that has 128 hidden nodes in each layer and used the *sigmoid* for the activation function. We compare the result with/without batch normalization, and deconvolution, where we remove the correlation between hidden nodes. Indeed, applying deconvolution to MLP networks outperforms batch normalization, as shown in Fig. 5(c,d).

# 7 CONCLUSION

In this paper we presented network deconvolution, a novel decorrelation method tailored for convolutions, which is inspired by the biological visual system. Our method was evaluated extensively and shown to improve the optimization efficiency over standard batch normalization. We provided a thorough analysis of its performance and demonstrated consistent performance improvement of the deconvolution operation on multiple major benchmarks given 10 modern neural network models. Our proposed deconvolution operation is straightforward in terms of implementation and can serve as a good alternative to batch normalization.

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

## A  APPENDIX

### A.1  SOURCE CODE

Source code can be found at:

https://github.com/yechengxi/deconvolution

The models for CIFAR-10/CIFAR-100 are adapted from the following repository:

https://github.com/kuangliu/pytorch-cifar.

### A.2  GENERALIZATION TO SEMANTIC SEGMENTATION

To demonstrate the applicability of network deconvolution to different tasks, we modify a baseline architecture of DeepLabV3 with a ResNet-50 backbone for semantic segmentation. We remove the batch normalization layers in both the backbone network and the head network and pre-apply deconvolutions in all the convolution layers. The full networks are trained from scratch on the Cityscape dataset (with $2,975$ training images) using a learning rate of $0.1$ for 30 epochs with batch size 8. All settings are the same with the official PyTorch recipe except we have raised the learning rate from $0.01$ to $0.1$ for training from scratch. Here we report the mean intersection over union (mIoU) curves of standard training and deconvolution using a crop size of $480$. We achieved significantly improved training results (Fig. 6).

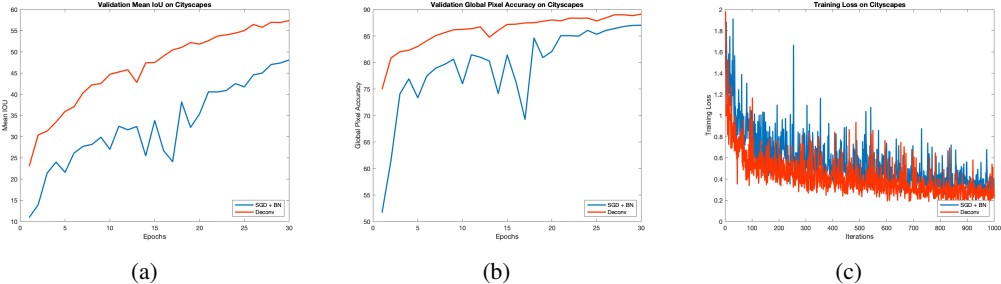

(a)  (b)  (c)

Figure 6: The mean IoU, pixel-wise accuracy and training loss on the Cityscapes dataset using DeepLabV3 with a ResNet-50 backbone. In our setting, we modified all the convolution layers to remove batch normalizations and insert deconvolutions.

### A.3    COUPLED/UNCOUPLED NEWTON SCHULZ ITERATIONS

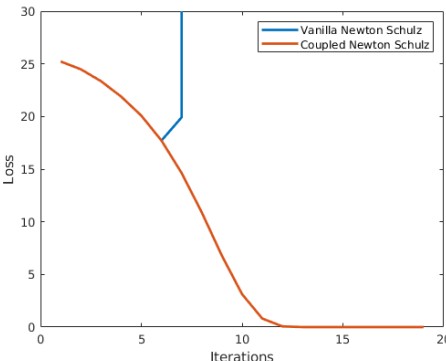

Figure 7: Comparison of coupled/uncoupled Newton-Schulz iterations on a $27 \times 27$ covariance matrix constructed from the Lenna Image.

We take the Lenna image and construct the $27 \times 27$ covariance matrix using pixels from $3 \times 3$ windows in 3 color channels. We apply the vanilla Newton-Schulz iteration and compare it with the coupled Newton-Schulz iteration. The Frobenius Norm of $D \cdot D \cdot Cov - I$ is plotted in Fig. 7. The rounding errors quickly accumulate with the vanilla Newton Schulz iterations, while the coupled iteration is stable. From the curve we set the iteration number to 15 for the first layer of the network to thoroughly remove the correlation in the input data. We freeze the deconvolution matrix $D$ after 200 iterations. For the middle layers of the network we set the iteration number to 5.

### A.4    REGULARIZATIONS

If two features correlate, weight decay regularization is less effective. If $X_1, X_2$ are strongly correlated features, but differ in scale, and if we look at: $w_1 X_1 + w_2 X_2$, the weights are likely to co-adapt during the training, and weight decay is likely to be more effective on the larger coefficient. The other, small coefficient is left less penalized. Network deconvolution reduces the co-adaptation of weights, and weight decay becomes less ambiguous and more effective. Here we report the accuracies of the VGG-13 network on the CIFAR-100 dataset using weight decays of $0.005$ and $0.0005$. We notice that a stronger weight decay is detrimental to the performance with standard training using batch normalization. In contrast, the network achieves better performance with deconvolution using a stronger weight decay. Each setting is repeated for 5 times, and the mean accuracy curves with confidence intervals of (+/- 1.0 std) are shown in Fig. 8(a).

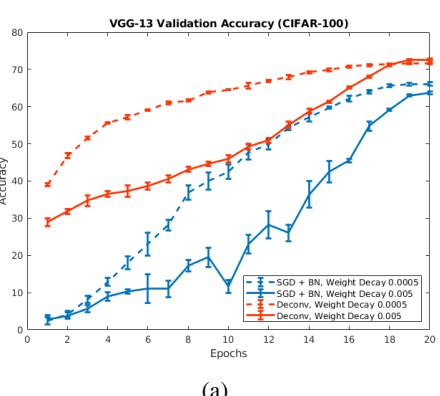 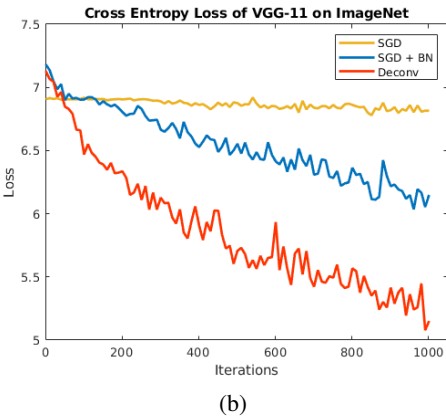

(a)                                     (b)

Figure 8: (a) The effects of weight decay on stochastic gradient descent (SGD) and batch normalization (BN) versus SGD and deconvolution (Deconv), training on the CIFAR-100 dataset on the VGG-13 network. Here we notice that increased weight decay leads to worse results for standard training. However, in our case with deconvolution, the final accuracy actually improves with increased weight decay (.0005 to .005). Each experiment is repeated 5 times. We show the confidence interval (+/- 1 std). (b)The training loss of the VGG-11 network on the ImageNet dataset. Only the first 1000 iterations are shown. Comparison is made among SGD, SGD with batch normalization and deconvolution.

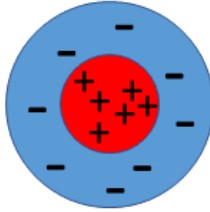 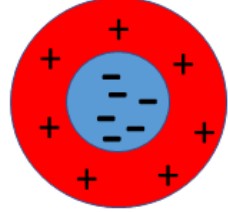

Figure 9: An on-center cell and off-center cell found in animal vision system. The on-center cell (left) responds maximally when a stimuli is given at the center and a lack of stimuli is given in a circle surrounding it. The off-center cell (right) responds in the opposite way.

### A.5   SPARSE REPRESENTATIONS FOR CONVOLUTION LAYERS

Network deconvolution reduces redundancy similar to the that in the animal vision system (Fig. 9). The center-surround antagonism results in efficient and sparse representations. In Fig. 10 we plot the distribution and log density of the signals at the first layer before and after deconvolution. The distribution after the deconvolution has a well-known heavy-tailed shape (Hyvrinen et al., 2009; Ye et al., 2013).

Fig. 11 shows the inputs to the 5-th convolution layer in $VGG - 11$. This input is the output of a $ReLU$ activation function. The deconvolution operation removes the correlation between channels and nearby pixels, resulting in a sharper and sparser representation.

### A.6   PERFORMANCE BREAKDOWN

Network deconvolution is a customized new design and relies on less optimized functions such as $Im2col$. Even so, the slow down is tunable to be around $\sim 10 - 30\%$ on modern networks. We plot the walltime vs accuracy plot of the VGG network on the ImageNet dataset. For this plot we use $B = 16, S = 4$ (Fig. 12). Here we also break down the computational cost on CPU to show deconvolution is a low-cost and promising approach if properly optimized on GPUs. We take random images at various scales and set the input/output channels to common values in modern networks.

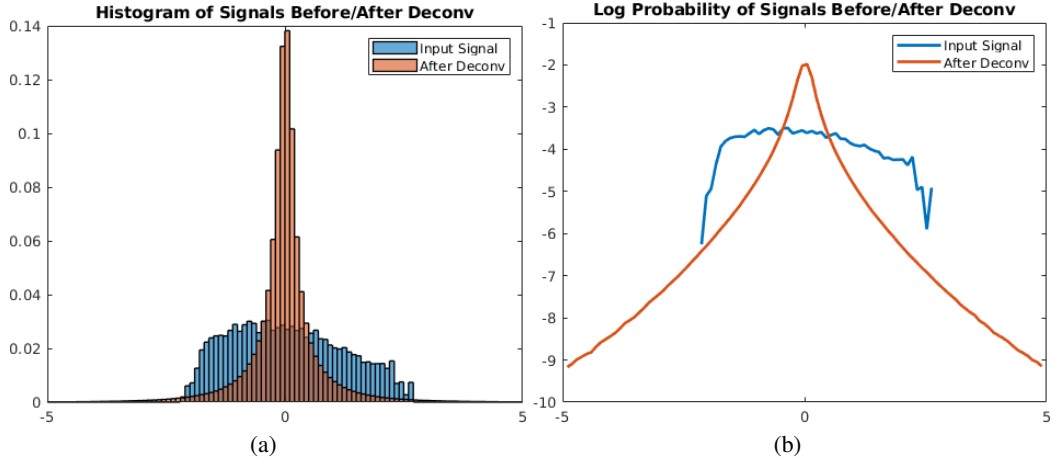

Figure 10: (a) Histograms of input signals (pixel values) before/after deconvolution. (b) Log density of the input signals before/after deconvolution.) The x-axis represents the normalized pixel value. After applying deconvolution, the resulting data is much sparser, with most values being zero.

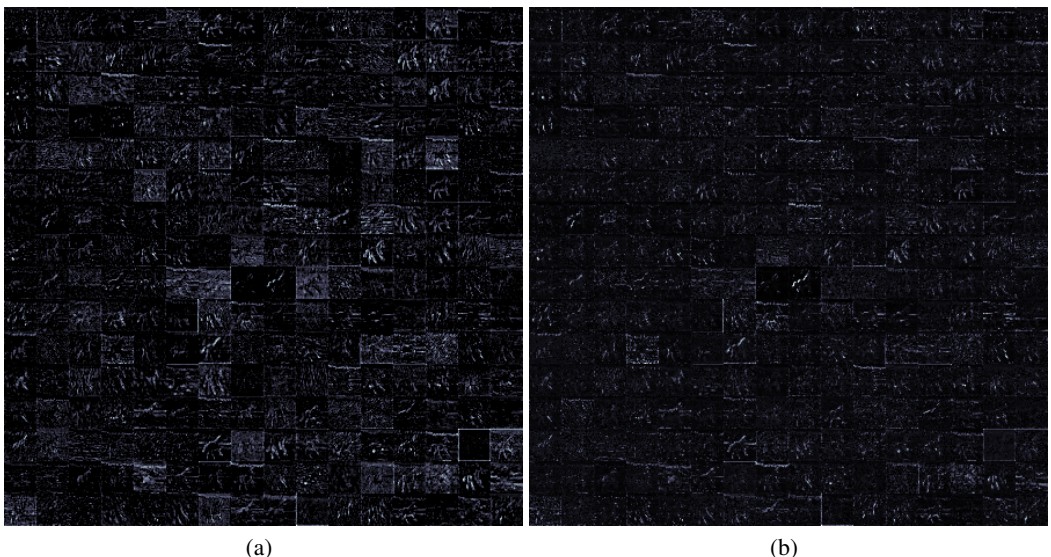

Figure 11: (a)Input features to the 5-th convolution layer in VGG-11. (b) Taking the absolute value of the deconvolved features. The features have been min-max normalized for visualization.(Best view on a display.)

The CPU timing on a batch size of 128 can be found in Table 3. Here we fix the Newton-Schulz iteration times to be 5.

## A.7 ACCELERATED CONVERGENCE

We demonstrate the loss curves using different settings when training the VGG-11 network on the ImageNet dataset(Fig. 8(b)). We can see network deconvolution leads to significantly faster decay in training loss.

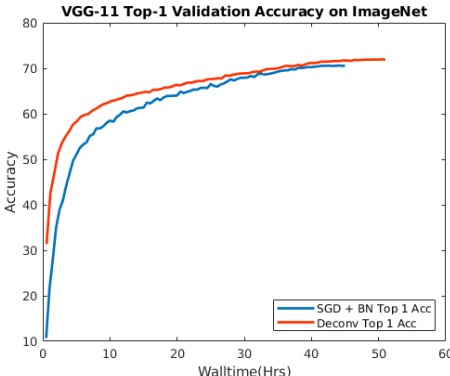

Figure 12: The accuracy vs walltime curves of VGG-11 networks on the ImageNet dataset using batch normalization and network deconvolution.

| $H$ | $W$ | $Ch_{in}$ | $Ch_{out}$ | $groups$ | $k$ | $stride$ | $Im2Col$ | $Cov$ | $Inv$ | $Conv$ |
|-----|-----|-----------|------------|----------|-----|----------|----------|-------|-------|--------|
| 256 | 256 | 3 | 64 | 1 | 3 | 3 | 0.069 | 0.0079 | 0.00025 | 0.50 |
| 128 | 128 | 64 | 128 | 1 | 3 | 3 | 0.315 | 0.3214 | 0.02069 | 0.67 |
| 64 | 64 | 128 | 256 | 2 | 3 | 3 | 0.045 | 0.0445 | 0.00076 | 0.55 |
| 32 | 32 | 256 | 512 | 4 | 3 | 3 | 0.022 | 0.0391 | 0.00222 | 0.48 |
| 16 | 16 | 512 | 512 | 8 | 3 | 3 | 0.011 | 0.0444 | 0.01155 | 0.23 |
| 128 | 128 | 64 | 128 | 64 | 3 | 3 | 0.376 | 0.0871 | 0.00024 | 0.66 |
| 64 | 64 | 128 | 256 | 128 | 3 | 3 | 0.042 | 0.0412 | 0.00082 | 0.53 |
| 32 | 32 | 256 | 512 | 256 | 3 | 3 | 0.023 | 0.0377 | 0.00208 | 0.47 |
| 16 | 16 | 512 | 512 | 512 | 3 | 3 | 0.011 | 0.0437 | 0.01194 | 0.22 |
| 128 | 128 | 64 | 128 | 32 | 3 | 3 | 0.360 | 0.0939 | 0.00021 | 0.67 |
| 64 | 64 | 128 | 256 | 32 | 3 | 3 | 0.044 | 0.0425 | 0.00076 | 0.55 |
| 32 | 32 | 256 | 512 | 32 | 3 | 3 | 0.023 | 0.0374 | 0.00204 | 0.46 |
| 16 | 16 | 512 | 512 | 32 | 3 | 3 | 0.011 | 0.0421 | 0.01180 | 0.22 |
| 256 | 256 | 3 | 64 | 1 | 3 | 5 | 0.030 | 0.0046 | 0.00029 | 0.49 |
| 128 | 128 | 64 | 128 | 1 | 3 | 5 | 0.153 | 0.1284 | 0.01901 | 0.66 |
| 256 | 256 | 3 | 64 | 1 | 7 | 3 | 0.338 | 0.1111 | 0.00107 | 0.69 |
| 256 | 256 | 3 | 64 | 1 | 7 | 5 | 0.180 | 0.0408 | 0.00107 | 0.69 |
| 256 | 256 | 3 | 64 | 1 | 7 | 7 | 0.063 | 0.0210 | 0.00104 | 0.70 |
| 256 | 256 | 3 | 64 | 1 | 11 | 3 | 0.681 | 0.4810 | 0.00479 | 0.99 |
| 256 | 256 | 3 | 64 | 1 | 11 | 5 | 0.315 | 0.1909 | 0.00488 | 1.00 |
| 256 | 256 | 3 | 64 | 1 | 11 | 7 | 0.199 | 0.1022 | 0.00494 | 0.99 |
| 256 | 256 | 3 | 64 | 1 | 11 | 11 | 0.069 | 0.0420 | 0.00496 | 1.00 |

Table 3: Breakdown of deconvolution layer component computation time (in sec., measured on CPU) against various layer parameters. The batch size is set to 128, $H \times W$ are layer dimensions, $Ch_{in}$ - number of input channels, $Ch_{out}$ - number of output channels, $groups$ - the number of channel groups, $k$ is the kernel size and $stride$ is the sampling stride.

## A.8   FORWARD BACKWARD EQUIVALENCE

We discuss the relation between the correction in the forward way and in the backward way. We thank Prof. Brian Hunt for providing us this simple proof.

Assuming that in one layer of the network we have $X \cdot W_1 = X \cdot D \cdot W_2 = Y$, $W_2 = D^{-1} \cdot W_1$, here $D = Cov^{-0.5}$.

$$\frac{\partial Loss}{\partial W_1} = \frac{\partial Loss}{\partial Y} \circ \frac{\partial Y}{\partial W_1} = X^t \cdot \frac{\partial Loss}{\partial Y}. \tag{6}$$

| BatchSize | Acc | LR | $\epsilon$ | Iter |
|---|---|---|---|---|
| 2 | 89.12% | 0.001 | 0.01 | 2 |
| 8 | 91.26% | 0.01 | 0.01 | 2 |
| 32 | 91.18% | 0.01 | 1e-5 | 5 |
| 128 | 91.56% | 0.1 | 1e-5 | 5 |
| 512 | 91.66% | 0.5 | 1e-5 | 5 |
| 2048 | 90.64% | 1 | 1e-5 | 5 |

Table 4: Performance and settings under different batch sizes.

Assuming $D$ is fixed,

$$\frac{\partial Loss}{\partial W_2} = \frac{\partial Loss}{\partial Y} \circ \frac{\partial Y}{\partial W_2} = (X \cdot D)^t \cdot \frac{\partial Loss}{\partial Y}. \tag{7}$$

One iteration of gradient descent with respect to $W_1$ is:

$$W_1^{new} = W_1^{old} - \alpha \frac{\partial Loss}{\partial W_1^{old}} = W_1^{old} - \alpha X^t \cdot \frac{\partial Loss}{\partial Y} \tag{8}$$

$\frac{\partial Loss}{\partial W_2} = \frac{\partial Loss}{\partial Y} \circ \frac{\partial Y}{\partial W_2} = (X \cdot D)^t \cdot \frac{\partial Loss}{\partial Y}.$

One iteration of gradient descent with respect to $W_2$ is:

$$W_2^{new} = W_2^{old} - \alpha \frac{\partial Loss}{\partial W_2^{old}} \tag{9}$$

$$D^{-1} \cdot W_1^{new} = D^{-1} W_1^{old} - \alpha \frac{\partial Loss}{\partial Y} \circ \frac{\partial Y}{\partial W_2^{old}} = D^{-1} W_1^{old} - \alpha (X \cdot D)^t \cdot \frac{\partial Loss}{\partial Y}. \tag{10}$$

We then reach the familiar form (Ye et al., 2017):

$$W_1^{new} = W_1^{old} - \alpha D^2 \cdot X^t \cdot \frac{\partial Loss}{\partial Y} = W_1^{old} - \alpha Cov^{-1}(\cdot X^t \cdot \frac{\partial Loss}{\partial Y}) \tag{11}$$

We have proved the equivalence between forward correction and the Newton's method-like backward correction. Carrying out the forward correction as in our paper is beneficial because as the neural network gets deep, $X$ gets more ill-posed. Another reason is that because $D$ depends on $X$, the layer gradients are more accurate if we include the inverse square root into the back propagation training. This is easily achievable with the help of automatic differentiation implementations:

$$\frac{\partial x_{i+1}}{\partial x_i} = D_{i+1} \circ f_i \circ W_i + \frac{\partial D_{i+1}}{\partial x_i} \circ f_i \circ W_i \tag{12}$$

Here $x_i$ is an input to the current layer and $x_{i+1}$ is the input to the next layer.

## A.9 INFLUENCE OF BATCH SIZE

We notice network deconvolution works well under various batch sizes. However, different settings need to be adjusted to achieve optimal performance. High learning rates can be used for large batch sizes, small learning rates should be used for small batch sizes. When the batch size is small, to avoid overfitting the noise, the number of Newton-Schulz iterations should be reduced and the regularization factor $\epsilon$ should be raised. More results and settings can be found in Table 4.

## A.10 IMPLEMENTATION OF FASTDECONV IN PYTORCH

We present the reference implementation in PyTorch. "FastDeconv" can be used to replace instances of "nn.Conv2d" in the network architectures. Batch normalizations should also be removed.

```python
import torch
import torch.nn as nn
import torch.nn.functional as F
from torch.nn.modules import conv
from torch.nn.modules.utils import _pair
import math

class FastDeconv(conv._ConvNd):
    def __init__(self, in_channels, out_channels, kernel_size, stride=1, padding=0, dilation=1,
groups=1, bias=True,
                 eps=1e-5, n_iter=5, momentum=0.1, block=64, sampling_stride=3, freeze=False,
                 freeze_iter=100):
        self.momentum = momentum
        self.n_iter = n_iter
        self.eps = eps
        self.counter = 0
        super(FastDeconv, self).__init__(
            in_channels, out_channels, _pair(kernel_size), _pair(stride), _pair(padding), _pair(
dilation),
            False, _pair(0), groups, bias, padding_mode='zeros')

        if block > in_channels:
            block = in_channels
        else:
            if in_channels % block != 0:
                block = math.gcd(block, in_channels)

        if groups > 1:
            # grouped conv
            block = in_channels // groups

        self.block = block
        self.num_features = kernel_size ** 2 * block
        if groups == 1:
            self.register_buffer('running_mean', torch.zeros(self.num_features))
            self.register_buffer('running_deconv', torch.eye(self.num_features))
        else:
            self.register_buffer('running_mean', torch.zeros(kernel_size ** 2 * in_channels))
            self.register_buffer('running_deconv', torch.eye(self.num_features).repeat(
in_channels // block, 1, 1))

        self.sampling_stride = sampling_stride * stride
        self.counter = 0
        self.freeze_iter = freeze_iter
        self.freeze = freeze

    def forward(self, x):
        N, C, H, W = x.shape
        B = self.block
        frozen = self.freeze and (self.counter > self.freeze_iter)
        if self.training:
            self.counter += 1
            self.counter %= (self.freeze_iter * 10)

        if self.training and (not frozen):
            # 1. im2col: N x cols x pixels -> N*pixles x cols
```

```python
            if self.kernel_size[0] > 1:
                X = torch.nn.functional.unfold(x, self.kernel_size, self.dilation, self.padding,
                                                self.sampling_stride).transpose(1, 2).contiguous(
)
            else:
                # channel wise
                X = x.permute(0, 2, 3, 1).contiguous().view(-1, C)[::self.sampling_stride ** 2,
:]

            if self.groups == 1:
                # (C//B*N*pixels, k*k*B)
                X = X.view(-1, self.num_features, C // B).transpose(1, 2).contiguous().view(-1,
self.num_features)
            else:
                X = X.view(-1, X.shape[-1])

            # 2. subtract mean
            X_mean = X.mean(0)
            X = X - X_mean.unsqueeze(0)
            self.running_mean.mul_(1 - self.momentum)
            self.running_mean.add_(X_mean.detach() * self.momentum)

            # 3. calculate COV, COV^(-0.5), then deconv
            if self.groups == 1:
                # Cov = X.t() @ X / X.shape[0] + self.eps * torch.eye(X.shape[1], dtype=X.dtype
, device=X.device)
                Id = torch.eye(X.shape[1], dtype=X.dtype, device=X.device)
                Cov = torch.addmm(self.eps, Id, 1. / X.shape[0], X.t(), X)
                deconv = isqrt_newton_schulz_autograd(Cov, self.n_iter)
            else:
                # Cov = X.transpose(1, 2) @ (X / X.shape[1]) + self.eps * Id
                X = X.view(-1, self.groups, self.num_features).transpose(0, 1)
                Id = torch.eye(self.num_features, dtype=X.dtype, device=X.device).expand(self.
groups, self.num_features,
                                                                                        self.
num_features)
                Cov = torch.baddbmm(self.eps, Id, 1. / X.shape[1], X.transpose(1, 2), X)

                deconv = isqrt_newton_schulz_autograd_batch(Cov, self.n_iter)

            # track stats for evaluation
            self.running_deconv.mul_(1 - self.momentum)
            self.running_deconv.add_(deconv.detach() * self.momentum)

        else:
            X_mean = self.running_mean
            deconv = self.running_deconv

        # 4. X * deconv * conv = X * (deconv * conv)
        if self.groups == 1:
            w = self.weight.view(-1, self.num_features, C // B).\
                    transpose(1, 2).contiguous().view(-1,self.num_features) @ deconv
            b = self.bias - (w @ (X_mean.unsqueeze(1))).view(self.weight.shape[0], -1).sum(1)
            w = w.view(-1, C // B, self.num_features).transpose(1, 2).contiguous()
        else:
            w = self.weight.view(C // B, -1, self.num_features) @ deconv
```

```python
            b = self.bias - (w @ (X_mean.view(-1, self.num_features, 1))).view(self.bias.shape)

        w = w.view(self.weight.shape)
        x = F.conv2d(x, w, b, self.stride, self.padding, self.dilation, self.groups)
        return x

def isqrt_newton_schulz_autograd(A, numIters, norm='norm', method='denman_beavers'):
    dim = A.shape[0]
    if norm == 'norm':
        normA = A.norm()
    else:
        normA = A.trace()

    I = torch.eye(dim, dtype=A.dtype, device=A.device)
    Y = A.div(normA)
    Z = torch.eye(dim, dtype=A.dtype, device=A.device)

    if method == 'denman_beavers':
        for i in range(numIters):
            # T = 0.5*(3.0*I - Z@Y)
            T = torch.addmm(1.5, I, -0.5, Z, Y)
            Y = Y.mm(T)
            Z = T.mm(Z)
    else:
        for i in range(numIters):
            # Z =  1.5 * Z - 0.5* Z@ Z @ Z @ Y
            Z = torch.addmm(1.5, Z, -0.5, torch.matrix_power(Z, 3), Y)
    # A_sqrt = Y* torch.sqrt(normA)
    A_isqrt = Z / torch.sqrt(normA)
    return A_isqrt

def isqrt_newton_schulz_autograd_batch(A, numIters):
    batchSize, dim, _ = A.shape
    normA = A.view(batchSize, -1).norm(2, 1).view(batchSize, 1, 1)
    Y = A.div(normA)
    I = torch.eye(dim, dtype=A.dtype, device=A.device).unsqueeze(0).expand_as(A)
    Z = torch.eye(dim, dtype=A.dtype, device=A.device).unsqueeze(0).expand_as(A)

    for i in range(numIters):
        T = 0.5 * (3.0 * I - Z.bmm(Y))
        Y = Y.bmm(T)
        Z = T.bmm(Z)
    # A_sqrt = Y*torch.sqrt(normA)
    A_isqrt = Z / torch.sqrt(normA)

    return A_isqrt
```