# OpenReview forum: "Network Deconvolution"
_ICLR.cc/2020/Conference — Accept (Spotlight)_

### Official Review · AnonReviewer2 · 2019-10-23
**Official Blind Review #2**

**Rating:** 8

**Review:**

This paper proposes "network deconvolution", a neural network primitive aimed at whitening the activations of each layer of the network. The method is a generalization of batch normalization that not only whitens per channel, but also removes correlations between channels and across spatial locations. Experiments show that the proposed methods improves training speed and predictive accuracy on a number of image classification models.

The method is novel and the proposed implementation details constitute a significant technical contribution. The experiments are not exhaustive and leave some open questions, but the first results are highly promising. The paper is clearly written, and embeds the proposed method in the literature.

I'd like to see more discussion and experiments on:
- What's the dependence on batch size? Does the method work better for larger batches? Or are small batches better for the added regularization effect, like what occurs with batch norm?
- Are results sensitive to the epsilon in algo 1? How is this chosen in practice and how does it interact with the approximate inversion of the covariance matrix?
- Under what conditions do you observe a speed-up in convergence? The only training curves shown in the main paper are on Fashion-MNIST which I don't think is very interesting. The appendix does offer a bit more information but I feel this deserves more attention.
- How does this method interact with regularization methods? For batch normalization much of the benefit seems to come from the noise it adds to network activations; is the same true for your method? Does network deconvolution still improve final accuracy if the baseline uses more extensive data augmentation or other regularization?

**Experience Assessment:**

I have published one or two papers in this area.

**Review Assessment: Checking Correctness Of Derivations And Theory:**

I assessed the sensibility of the derivations and theory.

**Review Assessment: Checking Correctness Of Experiments:**

I assessed the sensibility of the experiments.

**Review Assessment: Thoroughness In Paper Reading:**

I read the paper at least twice and used my best judgement in assessing the paper.

---

> ### Author Response · Authors · 2019-11-14
> **Re: Official Blind Review #2**
>
> Thank you for the attentive questions and experiment suggestions, we will try to answer each one to the best of our ability and also show some experiments that we ran in response to your suggestions.
>
> We have tested a wide range of batch sizes (2~2048). Under a learning rate 0.1, our method works best for batch sizes 128/256 on CIFAR10/ImageNet dataset. But it also works well with relatively small/large batch sizes. It does not seem to perform better by using a larger batch size as the number of iterations decreases. When the batch size is tiny, for example 4, we also need to reduce the learning rate to 0.01 to avoid the negative effects of noisy samples - this correction is necessary with batch normalization as well and is not unique to network deconvolution. More results can be found in Table 4. Another interesting observation we have is that we can start with a significantly larger learning rate (1.0) to train several networks, the final result can even be better than the most commonly used learning rate of 0.1. But this does not generalize to all architectures.
>
> Epsilon can make a small difference in the final result. We have set it to 1e-5, as in Batch Normalization. This value is fairly robust in the range between 1e-1 to 1e-5.
>
> The speedup in convergence seems universal across all networks and all datasets. The accuracy curve corresponding to deconvolution is higher than the curve corresponding to batch normalization if trained for the same number of iterations. Our updated Fig.7 also shows that not only the curves are better, but also the variations are smaller.
>
> In our opinion, the regularization effect of noise plays a less important role. Network deconvolution is a more optimal coordinate transform for correlated data. It makes the gradient to point more accurately to the minima. Data augmentation and regularization methods cannot achieve this goal.

---

### Official Review · AnonReviewer1 · 2019-10-23
**Official Blind Review #1**

**Rating:** 8

**Review:**

This paper proposes an operation for removing the pixel-wise and channel-wise correlations of input features. This operation can be considered as a generalization form of the former proposed decorrelated batch normalization. The approach has a well-sounded neurological inspired motivation and a solid explanation of the relationship with deconvolution. In the experimental analysis, the authors demonstrate a good performance of the proposed method compared with batch normalization. Also, the authors provide the CPU time of the proposed method, which is very appreciated.

Overall, this paper has reasonable contributions to the learning algorithm of the deep neural network, so I recommend the AC to accept this paper.

However, I have 2 negative points on this paper.
First, the computation cost for the im2col based convolution operation in a large kernel (eg. 7x7) is insanely large, and the authors only show us the results using VGG, which only has small 3x3 kernels.
Second, the arguments made on the sparse representations is somehow not convincing to me, it is really difficult to say the sparse representations has mad regularizations more effective with only 2 learning curves

**Experience Assessment:**

I have read many papers in this area.

**Review Assessment: Checking Correctness Of Derivations And Theory:**

I carefully checked the derivations and theory.

**Review Assessment: Checking Correctness Of Experiments:**

I carefully checked the experiments.

**Review Assessment: Thoroughness In Paper Reading:**

I read the paper thoroughly.

---

> ### Author Response · Authors · 2019-11-14
> **Re: Official Blind Review #1**
>
> Thank you for your in-depth review of the paper and the useful feedback.
>
> The use of large kernels has not been an issue for training on ImageNet for two reasons: (1) We can use a larger subsampling stride (e.g. 7) to compute the covariance matrix to amortize the cost of larger kernel size, so that each pixel will be used only once (Sec 4.5.2). We have also updated Table 3 in Sec A.5 to include runtimes of k=7,11.  (2) For the input layer, we only use the first 200 batches of images to calculate the running average deconvolution matrix. After that, we freeze the deconvolution matrix so that im2col is not called after that. This detail is also mentioned in Section A.2.
>
> Thank you for pointing the issue with our weight decay experiments; we have attempted to strengthen this result by re-running each setting for 5 times for the weight decay experiment in Sec A.3 replacing the curves with the mean accuracies with error bars over the runs.

---

### Official Review · AnonReviewer3 · 2019-10-25
**Official Blind Review #3**

**Rating:** 6

**Review:**

This paper addresses the correlation present in the input data, which may affect learning kernels with redundancy. Therefore, they introduce  a deconvolution mechanism on the input features to remove spatial and channel-wise correlation,  before these features are fed to network layers. They also show the deconvolution effect in the first layer resembles the center-surround effects found in biological neuron and induce sparsity in representation. They draw analogy to batch normalization, and show superiority in terms of  convergence and accuracy.

Overall, the concept of the paper is pretty simple and straightforward - basically it removes the correlation present in the input data, specifically in the case of convolution.The experimental results are promising and show fast convergence over using batch normalization, slightly better accuracy.

However, it seems that the deconvolution to avoid correlation is helping for classification tasks, but may not be applicable for other related tasks such as semantic segmentation, where simply using BN may work.

What is G in the overall complexity? No. of groups? Please define.

One of the questions that comes to mind is considering PCA normalization for the same purpose. Essentially, the step of computing the covariance is the same and decorrelating the data by  using the orthonormal basis vectors, pretty much with a similar motivation. Also, PCA is a linear transformation, which points to a way of learning in the network training. So, my question is what will be the problem of doing PCA transformation and then performing convolutions on that transformed decorrelated data? Maybe layerwise PCA transformation may also help in performance and reduce complexity? I think this discussion should be added to the normalization and whitening section of related works.

Minor note: The recommended page limit is 8 pages, the paper has now 10 pages. For me, Figure 1 is not conveying much information, and can be easily replaced with text. Also, I do not see the need of an extra section for the neurological basis for deconvolution; the brief discussion in the intro is enough.


**Experience Assessment:**

I have read many papers in this area.

**Review Assessment: Checking Correctness Of Derivations And Theory:**

I assessed the sensibility of the derivations and theory.

**Review Assessment: Checking Correctness Of Experiments:**

I assessed the sensibility of the experiments.

**Review Assessment: Thoroughness In Paper Reading:**

I read the paper at least twice and used my best judgement in assessing the paper.

---

> ### Author Response · Authors · 2019-11-14
> **Re: Official Blind Review #3**
>
> Thank you for taking the time to review, we will try our best to go down the line and answer each feedback question. We believe that deconvolution can be safely  coupled with convolution and applied for any tasks where convolution is used. To strengthen this argument,  in the Appendix Sec A.8 we have included a new experiment using FCN for semantic segmentation. In this experiment we have replaced the convolution layers in the FCN heads with deconv, while keeping the pretrained ResNet-50 backbone unmodified. We can also achieve improved training with this partial modification. There is room for larger improvements if the backbone network is trained from scratch, which can happen if pretraining is not available.
>
> We apologize for the error: the G in complexity is a typo. We have corrected it with B, the block size, thank you for pointing it out.
>
> The idea of using PCA is a good one, and one that we have put some thought into.  However, we came to the conclusion that PCA has a number of issues for whitening: 1. Finding the principle axes sequentially makes the algorithm slow. In addition, errors can accumulate as we solve for the smaller principle axes. 2. It is not well-defined if several axes have the same variance. Some extra discussion can be found in Hyvrinen et al., 2009 (Ch 5.2.3.2 and 5.2.4). Another similar  approach is Gram-Schmidt orthogonalization, however this approach doesn’t seem to work very well.

---

### Decision · Program_Chairs · 2019-12-19

**Decision:**

Accept (Spotlight)

**Comment:**

This paper presents a feature normalization method for CNNs by decorrelating channel-wise and spatial correlation simultaneously. Overall all reviewers are positive to the acceptance and I support their opinions. The idea and implementation is relatively straightforward but well-motivated and reasonable. Experiments are well-organized and intensive, providing enough evidence to convince its effectiveness in terms of final accuracy and convergence speed. Also, it’s analogy to biological center-surrounded structure is thought provoking. The novelty of the method seems somewhat incremental considering that there already exists a channel-wise decorrelation method, but I think the findings of the paper are interesting and valuable enough for ICLR community and would like to recommend acceptance.
Minor comments: I recommend authors to mention about zero-component analysis (ZCA) normalization, which has been a standard input normalization method for CIFAR datasets. I guess it is quite similar to the proposed method considering 1x1 convolution. Also, comparison with other recent normalization methods (e.g., Group Norm) would be useful.